# The Relationships between Sibling Characteristics and Motor Performance in 3- to 5-Year-Old Typically Developing Children

**DOI:** 10.3390/ijerph19010356

**Published:** 2021-12-30

**Authors:** Dagmar F. A. A. Derikx, Erica Kamphorst, Gerda van der Veer, Marina M. Schoemaker, Esther Hartman, Suzanne Houwen

**Affiliations:** 1Department for Human Movement Sciences, University Medical Centre Groningen, University of Groningen, P.O. Box 196, 9700 AD Groningen, The Netherlands; m.m.schoemaker@umcg.nl (M.M.S.); e.hartman@umcg.nl (E.H.); 2Inclusive and Special Needs Education Unit, Faculty of Behavioural and Social Sciences, University of Groningen, Grote Kruisstraat 2/1, 9712 TS Groningen, The Netherlands; e.kamphorst@rug.nl (E.K.); g.j.van.der.veer@rug.nl (G.v.d.V.); s.houwen@rug.nl (S.H.)

**Keywords:** motor skills, social environment, early childhood, child development

## Abstract

Siblings have been hypothesized to positively impact the motor performance of children by acting as examples and by providing a safe environment, but they may also negatively impact motor performance because they could compete for the parent’s time and care. Therefore, this study investigates the relationship between several sibling characteristics and motor performance in 3- to 5-year-old children. The sample consisted of 205 3- to 5-year-old children (mean age 50.9 ± 10.0 months, 52.2% boys). The Movement Assessment Battery for Children-2 was used to measure motor performance. The sibling variable was operationalized as the number of siblings present, the age difference between a child and its siblings, and the birth order position of a child. The confounding variables that were taken into account were the age, sex, and behavioral problems of the child and maternal education level. None of the investigated sibling variables were related to the total motor performance of a child after controlling for possible confounding variables. The absence of any associations between the sibling variables and motor performance might be explained by the characteristics of the study sample, the possibility that there is no linear relationship, or the presence of still unknown moderating or mediating factors.

## 1. Introduction

Early childhood is a period of rapid motor development. More specifically, between the ages of 3 and 5 years, children acquire and refine motor skills that enable them to participate in activities of daily living and to interact with objects and other people [1]. Motor performance can be defined as the execution of an activity or task that requires voluntary control over movements of the joints and body segments, so that smooth and efficient movements can be produced to achieve the intended outcome [2,3]. Motor performance can be further categorized as gross motor performance, which involves the coordination of large muscle groups, and fine motor performance, which involves movements made by smaller muscle groups and requires a high degree of precision [3,4].

Motor performance depends both on the genetic potential of a child and on the interaction of a child with the environment [5]. The environment in which a child grows up can be divided into the physical environment, which refers to all physical components of the area in which a child grows up, and the social environment, which includes all social interactions a child may have [6]. According to the ecological systems theory, this social environment is divided into four layers, which are categorized from the smallest layer of closest relationships with a child to the largest and outermost layer in a child’s environment [7]. The most influential environment of a child is called the microsystem, which consists of the closest relationships a child has, such as with family and the people within their home situation and day-care [7].

This closest and most influential social environment of a child also encompasses siblings. If children have siblings, they interact the most with their siblings, even more than with their parents and thus the presence of siblings may play an important role in the motor development of 3- to 5-year-old children [8,9]. Siblings may influence the development of young children in different ways. They may positively impact on motor development by serving as models and by providing a safe social environment in which motor skills can be practiced and developed [5,8]. However, the presence of siblings may also negatively affect the motor development when siblings act as competitors for parental resources [8]. More specifically, siblings may dilute the parent’s time and care available per child thereby decreasing opportunities of interaction and play between child and parent, which may have negative implications for a child’s motor development [8].

Despite the important role siblings have been theoretically argued to play in the development of motor skills in 3- to 5-year-old children, two recent studies investigating the relationship between the presence of siblings and fine and gross motor performance in 3-year-olds [10] and in 5- and 6-year-olds [11] found no significant relationships. More specific characteristics with regard to siblings such as the number of siblings and the birth order position of a child were investigated as well but resulted in mixed findings. No relationships were found between the number of siblings and gross motor performance [12,13,14], while mixed results were found for fine motor performance of 3- to 5-year-olds with one study finding no relationship [12], another study finding a relationship with only one out of five fine motor tasks measured [9], and a third study finding that the number of siblings predicted fine motor performance [13]. With regard to birth order position, one study reported that firstborn children performed better on fine and gross motor performance [12], another study stated that children with older siblings outperformed the firstborn children on gross motor performance [15], and the third study found no relationship between birth order position of a child and fine, gross, and total motor performance [16].

In conclusion, the scarceness of studies investigating the relationship between several characteristics regarding siblings and motor performance and the conflicting results emphasize the need for further investigation. Due to the rapid development of motor skills during the preschool years [1] and the suggested importance of the environment in this development [6], it is important to identify risk- or enabling factors such as the presence or absence of siblings so that children at risk of motor problems or delays can be recognized and monitored. This is especially relevant, as motor performance during early childhood plays an important role in other developmental domains such as the perceptual, cognitive, and social development [6,17]. Furthermore, motor performance has shown to be predictive for developing a physically active lifestyle [18,19] and academic achievement [20]. Knowledge regarding the relationship between several characteristics regarding siblings and motor performance can be used by parents, teachers, and health professionals to create stimulating situations involving siblings in which motor performance can be practiced and improved. Therefore, the aim of this study is to investigate the relationship between several characteristics regarding siblings and motor performance in 3- to 5-year-old children while controlling for possible confounding variables. Due to the large amount of time children spend with their siblings, a relationship between the sibling characteristics and motor performance is expected. However, whether this relationship will be positive or negative cannot be predicted yet, as siblings may have both a positive and a negative impact on motor performance.

## 2. Materials and Methods

### 2.1. Participants

This study was part of a larger longitudinal research project, examining motor performance, executive functioning, and language skills in Dutch-speaking 3- to 5-year-old children [21]. Participants were recruited from day-care centers, preschools, primary schools, and through public advertisements, social media, and snowball sampling. A parent-reported socio-demographic questionnaire was used to ascertain the absence of physical disabilities, neurological disorders (e.g., intellectual disability or autism spectrum disorder), and sensory impairments. The final sample consisted of 205 children (52.2% boys) with a mean age of 50.9 months (*SD* = 10.0 months, range = 36–71 months). This sample included 3-year-olds (*n* = 85, mean age = 41.1, *SD* = 3.4, 55.3% boys), 4-year-olds (*n* = 66, mean age = 52.9, *SD* = 3.5, 53.0% boys), and 5-year-olds (*n* = 54, mean age = 64.1, *SD* = 3.8, 46.3% boys).

### 2.2. Measurement of Study Variables

#### 2.2.1. Motor Performance

Motor performance was measured with age band 1 from the Movement Assessment Battery for Children 2 (MABC-2), Dutch Version [22]. This test battery assesses three subdomains of motor performance: manual dexterity (three items), aiming and catching (two items), and static and dynamic balance (three items). The raw scores were corrected for age and converted into a total standard score. The MABC-2 is a widely used test to measure motor performance in 3- to 5-year-old children. In this study, the translated Dutch version was used, which has its own norm scores. The Dutch version of the MABC-2 age band 1 has shown an excellent interrater reliability (Kappa ranging from 0.95 to 1.00) for the total standard test score [23], acceptable to good internal consistency [24], and a high construct validity, which was shown by a strong correlation with the Peabody Developmental Motor Scales [25].

#### 2.2.2. Siblings

In a socio-demographic questionnaire, parents were asked how many children were present in their household and what the ages of these children were in years. The variable siblings was operationalized as the number of siblings present, the age difference between a child and its siblings, and the birth order position of a child. Age difference was calculated in years between a child and its closest sibling. Birth order position was classified into five groups: being the youngest child, being the middle child, being the oldest child, being part of a twin, or being an only child.

#### 2.2.3. Confounding Variables

Child characteristics that were taken into account consisted of age, sex, and behavioral problems. Age and sex of a child were obtained with the socio-demographic questionnaire. Behavioral problems were assessed with the total difficulties score of the Strengths and Difficulties Questionnaire (SDQ) 2–4 for 3-year-old children, and with the SDQ4–17 for 4- and 5-year-old children [26]. The total difficulties score was generated by summing four subscales (i.e., emotional symptoms, conduct problems, hyperactivity, and peer problems scales) that consist of 5 items each. A higher score represents more behavioral problems. The total difficulties score of the SDQ shows acceptable psychometric properties in 3- to 5-year-old children [27,28,29].

The environmental confounding variable that was controlled for was maternal education level, which was obtained with the socio-demographic questionnaire and was used as a proxy for socioeconomic status (SES). Parental education level, income, and occupation are the three most used indicators of SES, with maternal education level being one of the strongest indicators of SES in child development research [30]. Maternal education was categorized as low (i.e., primary school and lower secondary education), intermediate (i.e., intermediate vocational level, higher secondary school, and pre-university education), or high (i.e., higher vocational education and university).

### 2.3. Procedure

The study received approval from the Ethics Review Committee of the Department of Pedagogical and Educational Sciences, Faculty of Behavioural and Social Sciences, University of Groningen. After given written information, parents were asked to sign an informed consent form before the assessments were performed. The data in this project were collected between April 2016 and May 2019. The participants were assessed by extensively trained assessors during two home-based sessions, which occurred on two non-consecutive days and lasted approximately 90–120 min each. While the parents filled out questionnaires, children completed a battery of tasks measuring motor performance, executive functioning, language, and general cognitive ability. The measurement of motor performance took approximately between 25 and 45 min per child. After each completed task, the children received a sticker, and after each assessment day, the children received a small gift. All assessments were videotaped for scoring purposes, and all data were stored using a personal study identifier to ensure confidentiality.

### 2.4. Data Analysis

All analyses were performed in IBM SPSS Statistics for Windows, Version 26 (IBM Corp., Armonk, NY, USA) and the significance level for all analyses was set at 0.05. First, missing data were dealt with by using multiple imputation. In order to create as precise and replicable imputation datasets as possible, several guidelines were followed [31]. Multiple imputation was only performed if the data were missing either at random or completely at random. Whether data were missing completely at random was tested with a Little’s MCAR test, with a non-significant result indicating that the data were missing completely at random. If this was not the case, it was checked whether data were missing at random by looking at the missing data patterns and theorizing whether these patterns were due to other observed variables or due to missing data itself. If the latter was the case, then the data were not missing at random, and multiple imputation could not be performed. Participants with more than 50% of missing data on the variables used in this study were excluded from the dataset. Multiple imputation was performed for all variables that had between 5% and 50% of missing data, while all variables included in this study were used as auxiliary variables [31]. The number of imputations depended on the percentage of missing data, with 20 imputations being used for up to 30% of missing data and 40 imputations being used for between 30% and 50% of missing data [32].

To investigate the relationship between the sibling characteristics and motor performance, bivariate correlations and a hierarchical regression analysis were performed. The bivariate correlations between the key and confounding variables were assessed with Pearson Correlations when both variables were continuous and with point biserial correlation when one variable was dichotomous. A correlation coefficient was considered weak between 0.1 and 0.3, moderate between 0.3 and 0.5 and strong between 0.5 and 1.0 [33]. A hierarchical regression analysis was performed to investigate whether the sibling variables (i.e., number of siblings, age difference, and birth order position) accounted for incremental variance in the MABC-2 Total Score after controlling for possible confounding variables (i.e., age, sex, behavioral problems, and maternal education level). The required sample size was determined using the formula: N > 50 + 8 * number of independent variables [34,35,36]. This is a general rule of thumb that has often been used to calculate the required sample size for multiple regression analyses. The assumptions of normality of residuals, linearity, homoscedasticity, and no multi-collinearity were tested. Multi-collinearity was indicated by tolerance values below 0.1 and variance inflation factor (VIF) values above 10 [37]. Due to the multiple imputation, both the tests to check assumptions and the analyses were performed on all the imputed datasets separately. Assumptions were assumed to be violated if tests showed that assumptions were not met in more than 20% of the imputation datasets. In the first step of the hierarchical regressions, the confounding variables age, sex, behavioral problems, and maternal education level were entered blockwise, and in the second step, the number of siblings, age difference, and birth order position were added blockwise. The categorical variable maternal education level was entered as a dummy variable whereby high maternal education level was taken as a reference category, since this category included the largest fraction of the cases. Birth order position was also entered as a dummy variable, and the category ‘only child’ was taken as the reference value, so that children with siblings could be compared to children without siblings. IBM SPSS Statistics for Windows, Version 26 (IBM Corp., Armonk, NY, USA) supports pooling some of the results such as the *B*-value, standard error (*SE*), *t*-value, and *p*-value. However, the ß, *R*^2^, *R*^2^ change, and significance of the *R*^2^ change could not be pooled by SPSS and therefore were manually pooled by averaging the results from the 20 imputation datasets as has been recommended by previous research [38,39].

## 3. Results

### 3.1. Preliminary Analyses

The final sample consisted of 205 children, after four children (1.9%) were excluded from the dataset because more than 50% of the variables were missing for these children. Multiple imputation was performed for the total difficulties score of the SDQ, the total score of the MABC-2, and the age difference between a child and its sibling, as these variables had between 5% and 50% of missing data. The data on these variables were missing completely at random (Little’s MCAR test: χ^2^ (25) = 32.48, *p* = 0.145). Based on the percentages of missing data, 20 imputations were created [32]. Descriptive statistics for the study variables including the pooled mean and the mean and range of the standard deviations from the 20 imputation datasets are presented in Table 1. The assumption of no outliers was checked for all continuous variables and revealed 35 outliers in the total difficulties score of the SDQ or in the age difference between a child and its siblings across the 20 imputation datasets. These outliers were adjusted to a score corresponding to a *z*-score of ±3.29 [40], which was calculated per variable and per imputation dataset. The other assumptions of normality of residuals, linearity, homoscedasticity, and no multi-collinearity were met.

### 3.2. Bivariate Correlations

Bivariate correlations were calculated between the key and confounding variables in the 20 imputed datasets (Table 2). The age and sex of a child were weakly correlated with total motor performance, which indicated that younger children and boys performed better on the age-standardized total score of the MABC-2 than older children and girls. The total difficulties score of the SDQ was weakly negatively related to being the youngest child and total motor performance and was weakly positively related to being part of a twin. This indicates that being the youngest child and having better motor performance were related to less behavioral problems, while being part of a twin was related to more behavioral problems. In addition, none of the sibling variables were correlated with the motor performance of a child.

### 3.3. Regression Analyses

The formula used to calculate the required sample size (i.e., N > 50 + 8 * number of independent variables [34,35,36]) revealed a required sample of 138 participants for the hierarchical multiple regression analysis to investigate the relationship between the sibling characteristics and motor performance (number of predictors = 11). This required sample size was obtained. The hierarchical regression analyses showed that the separate siblings variables did not significantly explain any additional variance in the total motor score. The total model did not significantly explain any additional variance either for the total motor score after controlling for age, sex, behavioral problems, and maternal education level (Table 3).

## 4. Discussion

### 4.1. Main Findings

The aim of this study was to investigate the relationship between several characteristics regarding siblings and motor performance of 3- to 5-year-old children after controlling for possible confounding variables. A relationship between the sibling characteristics and motor performance was expected, due to the large amount of time children and siblings spent together. However, the direction of this relationship could not be predicted beforehand, because both a positive and a negative impact of siblings on the motor performance of a child has been described in previous literature [5,8]. Siblings may positively impact motor performance by acting as examples and by providing a safe environment in which motor performance can be practiced [5,8], while they may also negatively impact motor performance because they could compete with a child for their parent’s time and care [8]. Contrary to the hypothesis of finding a relationship, none of the investigated sibling variables, which encompassed the number of siblings present, the age difference between a child and its siblings, and the birth order position of a child, were related to the total motor performance of a child after controlling for possible confounding variables.

Prior studies investigating the relationship between sibling characteristics and motor performance have mainly focused on the presence or number of siblings [9,10,11,12,13,14] and birth order position of a child [12,15,16], while to the best of our knowledge, the age difference between a child and its siblings has not been investigated yet. The studies focusing on the presence or number of siblings found no relationships with gross motor performance of 3- to 5-year-olds [10,11,12,13,14], while mixed results were found for the fine motor performance of 3- to 5-year-olds [9,12,13]. One study found no relationship [12], another study found a relationship with only the task ‘copying a square’ out of the five fine motor tasks measured [9], while a third study found that the number of siblings was a predictor of better fine motor performance [13]. Our results regarding gross motor performance were consistent with these findings in that no significant relationships were found with the sibling variables. Our results regarding fine motor performance were in line with the studies that found no or hardly any relationship with any of the sibling variables [9,12]. The conflicting results found in the previously conducted studies and the current study regarding fine motor performance might be explained by the confounding variables used. The study that did find a relationship controlled for several confounding variables including parental education level, maternal perceived pressure, family support, and mental health [13], whereas the studies that found no or hardly any relationship, including the current study, included different confounding variables or did not include any confounding variables at all [9,12]. The specific confounding variables included in the study that found a relationship [13] may be related to the negative impact siblings might have on motor performance, such as the increased pressure on parents and reduced resources per child. One hypothesis could be that siblings simultaneously have a negative and a positive impact on motor performance, which thereby cancel each other out. By controlling for the negative impact siblings might have, the positive impact siblings might have on the motor performance of a child may have been revealed in the previous study [13]. In conclusion, no relationship has been found between the presence or number of siblings and gross motor performance [10,11,12,13,14], and only one study found the presence of siblings to be related to better fine motor performance [13], while the rest of the previous studies and the current found no or hardly any relationships [9,12]. These findings start to suggest that there is no relationship between the presence or number of siblings and motor performance. The confounding variables used may play a big role in further unravelling these relationships, and therefore, confounding variables controlling for either the negative or the positive impact siblings might have on motor performance should be further investigated so that both the negative and the positive impact can be examined separately.

Prior studies investigating the relationship between the birth order position and motor performance of 3- to 5-year-old children resulted in mixed findings as well, with one study reporting that firstborn children performed better on fine and gross motor performance [12], another study stating that children with older siblings outperformed the firstborn children on gross motor performance [15], and the third study finding no relationship between birth order position of a child and fine, gross, and total motor performance [16]. According to previous literature, the birth order position of a child might impact motor performance either positively or negatively. Children with older siblings might observe and imitate the developmentally more advanced motor performance of their older siblings, which might positively impact their motor performance [5,8]. However, firstborn children have access to the undivided resources of their parents in the beginning of their life, which might positively impact their motor performance [8]. In our study, no significant relationship was found in any direction, and it is not yet clear why our study and these previous studies led to such mixed outcomes. Since the findings are so mixed, no conclusions can be drawn yet, and further research is needed to discover whether there is a relationship between birth order position and motor performance and if so, in which direction. Confounding variables controlling for either the negative or the positive impact siblings might have on motor performance should be included in these future studies as well.

Although a relationship between the sibling variables and motor performance was absent, it was striking that we found some significant relationships between the sibling and motor performance variables on the one hand and the total difficulties score of the SDQ on the other hand. Being the youngest child and having better motor performance were related to less behavioral problems, while being part of a twin was related to more behavioral problems. These findings indicate that the birth order position of a child is related to the amount of behavioral problems and that in turn, the amount of behavioral problems is related to the motor performance of a child. A previous study investigated the relationship between the presence of siblings on the one hand and motor skills and socio-emotional skills on the other hand [41]. This study reported that the presence of siblings was not related to motor skills, while children with siblings scored higher on social relations (i.e., being able to establish relationships, sharing with others, showing respect toward others, and showing both positive and negative emotions while playing with others) [41]. In other words, having siblings to play with during childhood might train a child in establishing social relationships with others and how to behave during these interactions. This was also shown in another study that reported that children with siblings were more popular and accepted by their peers [42] and was also confirmed by our finding that youngest children show fewer behavioral problems. However, the contrasting result that twins show more behavioral problems was also found, indicating that having a sibling of the same age to play with is disadvantageous for the behavior of a child. This may be explained by the challenging situation it may create for parents when having twins. Parents of twins generally experience more stress, financial troubles, and decreases in marital satisfaction than parents of singletons [43], which could be associated with higher levels of problem behaviors in their children. In addition to the relationships between the sibling variables and behavioral problems, it was also found that having fewer behavioral problems was related to having better motor performance. This relationship may be bidirectional [44]. Children may improve their motor skills through interaction with peers by observing and imitating their motor skills and practicing their own motor skills. However, the level of motor skills might also influence how children interact with peers [45,46]. Overall, we can assume that children with older siblings have less behavioral problems, and this is in turn related to having better motor performance. This may indicate an indirect relationship between the presence of siblings and motor skills, although this indirect relationship was not found in the present study. An explanation for this might be that this indirect relationship is not yet applicable in this age group and becomes increasingly more important as the child grows older and the social environment expands as well [47].

The absence of any associations between the sibling variables and motor performance found in the current study might be due to the characteristics of the study sample such as SES and the school age of the children. On the one hand, siblings might negatively impact the motor performance by diluting parental resources [8]. It has been suggested that in families of lower SES, fewer resources such as time and money are available, and thus, siblings act as competitors for parents’ time and care, thereby hindering the development of a child instead of supporting it [48]. However, in our study, in which mainly families of higher SES (78.0%) were included, these negative effects might not have been found due to the high amount of available resources that could be divided. On the other hand, previously found positive relationships between the presence of siblings and motor performance were explained by the safe social environment siblings might provide in which motor performance can be practiced and developed [5,8]. However, the social environment of a child includes other people besides siblings, for example parents and peers. In the Netherlands, children start to attend primary school from the age of four, and prior to that, many children attend preschool and/or day-care. This factor was not controlled for in our study, but attending preschool, day-care, and primary school gives children the opportunity to interact with peers and learn and imitate their behavior [49,50]. This may diminish the effect siblings might have on 3- to 5-year-old children, or rather, it might compensate for the absence of siblings.

Another possible explanation for not finding any significant relationships might be that there is no linear relationship between the sibling variables and motor performance. The development of a child is influenced by factors within the individual, the task, and the environment [51], which operate as an integrated whole. If these factors are studied in isolation, as is mainly done in previous literature, they may lose their impact, because all these elements interact and influence development in a complex and integrated fashion [52,53]. A variable-centered approach, such as used in many studies as well as this study, assumes that all individuals in a sample are drawn from a single population, that the relationships between all variables are linear, and that all relationships are similar for all individuals [54]. However, since all children have their ‘own set’ of individual, task, and environmental constraints [53] and motor skills have been suggested to develop in a non-linear manner [55], a person-centered approach should be considered in future research. A person-centered approach considers the possibility that a sample might include several subpopulations, which are characterized by various constellations of parameters [53]. One such parameter might be the entire social environment of the child. Every child has its own social environment, which is composed of actors such as parents and peers. The influence of interaction with the various actors within the social environment is not identical due to the different type of relationships, but it is rather synergistic, as less interaction with one actor might be compensated by more interaction with another actor [48]. For example, children without siblings might compensate this by looking for more interaction with parents or peers. By using a person-centered approach, subgroups and individual profiles regarding for example the social environment composition can be identified to further unravel the relationship between the sibling variables and motor performance [56].

Another possible explanation for not finding a significant relationship might be that the relationship between the sibling variables and motor performance is different than expected in the current study. Other, possibly still unknown, moderating, mediating, or covariate factors may play a role in this relationship. As mentioned previously, the age at which children attend day-care, preschools, and primary school may moderate the relationship between sibling variables and motor performance. In addition, previous research has shown that the covariate quality of schooling at a young age can influence motor performance enormously [5,57,58], and therefore, this factor should be taken into account as well. Another variable that may mediate the relationship between the sibling characteristics and motor performance is parenting practices. Siblings influence parenting practices by providing learning opportunities and by serving as social comparison for parents [59], which in turn may impact the motor performance of children [60]. Furthermore, additional sibling variables should be studied to further elucidate which sibling characteristics are related to the motor performance of children and what the exact nature of this relationship is. According to the social learning theory, children are more likely to model the behavior of siblings who they are more similar to or more intimate with [59,61]. Therefore, variables such as the sex of the sibling and the quality of the relationship between a child and its sibling should be taken into account as well. Another sibling characteristic that might influence how siblings affect the development of a child is the sibling’s proficiency in motor performance. Siblings may affect the development of a child by acting as a model. However, if the siblings’ levels of proficiency in motor performance are insufficient to model, a child will not gain much by imitating these motor skills [62].

### 4.2. Strengths and Limitations

The most important strength of our study is the large sample of included children with an equal sex distribution and a narrow age range. Furthermore, in contrast to previous studies, this study investigated multiple sibling variables including the age difference between a child and its siblings which, to the best of our knowledge, has not been investigated before. However, this study also has some limitations that need to be considered. Maternal education level, which was used as an indicator for SES, was unequally distributed with mainly highly educated mothers being included in our study. Therefore, the finding of this study might not be generalizable to the whole population, as the SES of the included sample was higher than that of the average population. Another important point to address is the validity of the assessments. Assessing performance-based measured in children is a challenge due to factors such as fatigue, limited attention span, motivation, and confidence of a child [63]. However, by skillfully engaging and motivating the children and by planning breaks in the assessments, the influence of these factors on the assessments was minimized.

## 5. Conclusions

Contrary to the expectations, none of the investigated sibling variables, which included the number of siblings present, the age difference between a child and its siblings, and the birth order position of a child, were related to motor performance. Our findings in combination with previous findings suggests that there is no relationship between the presence or number of siblings and motor performance, whereas for the birth order position of a child, the findings are too mixed to draw any conclusions. The absence of any associations between the sibling variables and motor performance found in the current study might lie in the characteristics of the study sample, which consisted of children with a relatively high SES who attended school at a relatively young age. However, other possible explanations for the absence of any significant relationship between the sibling variables and motor performance might be that there is no linear relationship or that there are other, still unknown, moderating or mediating factors that should be taken into account. More research, which also includes confounding variables that control for either the negative or the positive impact siblings might have on motor performance, is required to further elucidate which sibling characteristics are related to motor performance and what the exact nature of this relationship is. In addition, further research is required into the possible indirect relationship between the presence of siblings and motor skills mediated by behavior and the behavioral problems of a child.

## Figures and Tables

**Table 1 ijerph-19-00356-t001:** Pooled means and standard deviations for the study variables and percentage of missing data before multiple imputation.

	Percentage	Mean	*SD* (Mean (Range))	Missing Data Before Multiple Imputation (%)
*Confounding variables*				
Age (months)		50.91	10.00	0.0
Sex	52.2			0.0
Total difficulties score (SDQ)		7.46	4.67 (4.57–4.71) ^b^	5.4
Maternal education level				2.4
Low	3.0			
Intermediate	19.0			
High	78.0			
*Siblings*				
Number of siblings				2.4
0	12.0			
1	53.0			
2	27.0			
3	5.5			
4	1.5			
5	1.0			
Age difference (years)		2.06	1.51 (1.44–1.64) ^b^	7.8
Birth order position				4.9
Only child	12.3			
Youngest child	32.3			
Middle child	13.3			
Oldest child	36.9			
Twin	5.1			
*Motor performance*				
MABC-2 Total score ^a^		9.54	3.34 (3.23–3.44) ^b^	15.6

*Note.* Sex: 0 = girls, 1 = boys; SDQ = Strengths and Difficulties Questionnaire Hyperactivity-Inattention subscale; MABC-2 = Movement Assessment Battery for Children-2. ^a^ Standardized for age. ^b^ Mean and range of standard deviations for the 20 multiple imputation datasets.

**Table 2 ijerph-19-00356-t002:** Bivariate correlation matrix for the key and confounding variables.

	2	3	4	5	6	7	8	9	10	11	12
1.Age (months)	0.02	0.01	−0.04	0.05	0.01	0.12 *	−0.05	−0.05	0.20 **	−0.09	−0.17 *
2. Sex		−0.16 *	0.01	0.07	0.01	−0.05	0.03	−0.02	−0.06	0.05	0.21 **
3. SDQ			0.02	0.03	0.05	−0.06	−0.13 *	0.00	0.03	0.19 **	−0.14 *
4. Low maternal education				−0.09	0.07	−0.02	0.00	0.10	−0.14 *	−0.04	0.06
5. Intermediate maternal education					−0.09	0.00	0.04	−0.20 **	0.00	0.12	−0.11
6. Number of siblings						0.18 **	0.02	0.52 ***	−0.12	0.23 **	−0.09
7. Age difference							0.49 ***	−0.07	0.06	−0.33 ***	−0.03
8. Youngest child								−0.28 ***	−0.53 ***	−0.16 *	−0.02
9. Middle child									−0.30 ***	−0.09	−0.04
10. Oldest child										−0.18 **	0.05
11. Twin											−0.12
12. MABC-2 Total score											

*Note.* Sex: 0 = girls, 1 = boys; SDQ = Total Difficulties score of the Strengths and Difficulties Questionnaire; MABC-2 = Movement Assessment Battery for Children-2. * *p* < 0.05. ** *p* < 0.01. *** *p* < 0.001.

**Table 3 ijerph-19-00356-t003:** Hierarchical multiple regression analysis predicting the MABC-2 total score from sibling variables after controlling for confounding variables.

		MABC-2 Total Score				
Step	Predictors	*B*	*SE B*	β	*t*	*p*
1	Age (months)	−0.06	0.03	−0.17	−2.14	0.033 *
	Sex	1.36	0.50	0.21	2.73	0.006 **
	SDQ	−0.07	0.06	0.04	−1.25	0.214
	Low maternal education level	0.84	1.33	−0.11	0.64	0.526
	Intermediate maternal education	−0.94	0.63	−0.10	−1.49	0.137
	*R* ^2^	0.10				
	*F* change	4.36				
	*p* change	0.002 **				
2	Age	−0.06	0.03	−0.19	−2.31	0.022 *
	Sex	1.44	0.50	0.22	2.87	0.004 **
	SDQ	−0.06	0.06	0.05	−0.96	0.339
	Low maternal education	0.99	1.36	−0.11	0.73	0.463
	Intermediate maternal education	−0.93	0.65	−0.08	−1.42	0.155
	Number of siblings	−0.08	0.44	−0.02	−0.18	0.856
	Age difference	−0.04	0.23	−0.02	−0.19	0.850
	Youngest child	−0.28	1.30	−0.04	−0.21	0.831
	Middle child	−0.64	1.52	0.05	−0.42	0.676
	Oldest child	0.33	1.12	−0.07	0.30	0.768
	Twin	−1.79	1.95	−0.12	−0.92	0.361
	*R* ^2^	0.14				
	Δ*R*^2^	0.03				
	*F* change	1.18				
	*p* change	0.377				

*Note.* Age, sex, behavioral problems, and maternal education level were entered in the first step. The sibling variables were entered in the second step; Sex: 0 = girls, 1 = boys; SDQ = Total Difficulties score of the Strengths and Difficulties Questionnaire. * *p* < 0.05. ** *p* < 0.01.

## Data Availability

The data presented in this study are available on request from the corresponding author. The data are not publicly available because parental consent was only given for use of data by the researchers directly involved in the MELLE study.

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
