# Peer review of "The Relationships between Sibling Characteristics and Motor Performance in 3- to 5-Year-Old Typically Developing Children"

_ijerph, 2021, doi:10.3390/ijerph19010356_

Round 1
Reviewer 1 Report
Thank you for the opportunity to review the manuscript “The Relationships Between Sibling Characteristics and Motor Performance in 3- to 5-Year-Old Typically Developing Children”.
The authors have done excellent work recruiting a sample of 205 children from the Netherlands 3 to 5 years of age and using the Movement Assessment Battery for Children-2 (MABC-2) Dutch version to measure motor performance. The manuscript is very well written, the analyses and results are presented very well too.
Some very minor questions arose for this reviewer.
The authors mention on page 3 lines 110 to 115 that the MABC-2 is reliable and valid. Since this is the translated Dutch version, can the authors elaborate on the reliability, and validity of the Dutch version how well the measurement model holds up to the original measurement model derived by the developers. Next how well do the psychometric properties hold up with this sample?
The sentence from page 5 lines 211 to 213 needs some clarification “These outliers were adjusted to a score corresponding to a z-score of ± 3.29 [38], which was calculated per variable and per imputation dataset.” As it is not clear if this process was used for the MABC-2 scores as well as the other variables with missing data in the model
Last, what software was used for the sample size calculation? This needs to be cited.
Reviewer 2 Report
Thanks you for inviting me to review this paper which sought to understand the relationship between several sibling characteristics and motor performance in 3- to 5- 16 year-old children. I commend the authors for conducting such a complex study. In my opinion, there are few areas that needs to be strengthened before the paper could be published. I will encourage the authors to address them to strengthen their paper.
- When was the data collected and what was the duration of each data collection?
- I don't think power analysis was necessary since there is literature on adequate sampling for regression (please check Pallant, 2016... SPSS Survival manual)
- I have been wondering why the authors decided to add parental data in this study. I think this is a distraction and would recommend they work with only the sibling data. In this case, linear regression will be more ideal for this study.
- I think the discussion should be rewritten. The authors merely repeated their findings and other studies without interrogating the study findings. The authors position should be more apparent here.
- The conclusion could be strengthened and would recommend addition study implication for child development.
Reviewer 3 Report
It is a very well written and methodologically correct article, I only have one question and a couple of suggestions: why did the authors not consider other confounding variables such as the father's educational level? As suggestions I think you should cite this article: doi.org/10.3389/fped.2021.684418, it is a very recent article (published in July 2021) that can greatly enrich your discussion because the authors do not find differences between children who have siblings in physical-motor aspects, but they do find differences in perceptual and affective-relational aspects. This suggestion is closely related to my second recommendation: I am struck by the positive correlation between the scores on the SDQ and the twins, and the negative correlation between the SDQ and the MABC-2. I understand that, in your study, the SDQ was considered as a confounding variable, but in the discussion I do not find an explanation for this interesting result or how to incorporate it into future studies (perhaps, include the SDQ as a mediating / moderating variable ). The socio-emotional aspects are fundamental in this type of study.
Round 2
Reviewer 2 Report
I thank the authors for addressing my comments and offer explanation to where they disagreed with me. I will gladly recommend the paper for publication.